# The Productivity and Financial Impacts of Eight Types of Environmental Enrichment for Broiler Chickens

**DOI:** 10.3390/ani10030378

**Published:** 2020-02-26

**Authors:** Philip J. Jones, Fernanda M. Tahamtani, Ida J. Pedersen, Jarkko K. Niemi, Anja B. Riber

**Affiliations:** 1School of Agriculture, Policy and Development, University of Reading, P.O. Box 237, Earley Gate, Whiteknights, Reading, Berkshire RG6 6AR, UK; 2Department of Animal Science, Aarhus University, Blichers Allé 20, P.O. Box 50, DK-8830 Tjele, Denmark; fernandatahamtani@anis.au.dk (F.M.T.); anja.riber@anis.au.dk (A.B.R.); 3HK Scan, Tvaermosevej 10, 7830 Vinderup, Denmark; idajust28@gmail.com; 4Department of Veterinary and Animal Sciences, University of Copenhagen, 1870 Frederiksberg, Denmark; 5Natural Resources Institute Finland (Luke), Bioeconomy and Environment Unit, Kampusranta 9, FI-60320 Seinäjoki, Finland; jarkko.niemi@luke.fi

**Keywords:** Broiler flocks, health management, environmental enrichment, financial assessment

## Abstract

**Simple Summary:**

Fast growing broiler birds have an elevated risk of leg health problems through inactivity. Increasing the complexity (enriching) of the rearing environment, e.g., adding straw bales into broiler houses, is suggested as a way of increasing activity levels. While a number of studies have examined the impact of enrichments on bird activity levels and health, few have examined their financial impacts. This is problematic, because enrichments which cost money to implement that do not provide an obvious financial benefit are unlikely to be adopted without regulation. This study examines the financial impacts of eight enrichments, accounting for the cost of the enrichment and changes to both bird productivity, e.g., growth rates and market prices. The study found financial benefits from only one of the enrichments (increased distance between feed and water to 3.5 m) and financial losses in most cases, due to the costs of the enrichments. The impacts of the enrichments on bird productivity are relatively minor. The study suggests that if widespread adoption of these enrichments, to obtain welfare benefits, is to be achieved, some form of market incentive will need to be provided, such as a price premium paid by consumers in return for an enhanced rearing environment.

**Abstract:**

Reduced mobility in broilers can contribute to leg health problems. Environmental enrichment has been suggested as one approach to combat this through stimulating increased physical activity. Past studies have tested the effect of environmental enrichments on bird behaviour, health and welfare, but few have estimated their financial impacts. This study tested the impact of eight types of environmental enrichment on enterprise net margin, accounting for direct intervention costs plus indirect effects via changes to bird mortality, weight, feed intake, feed conversion ratio, and foot pad dermatitis. The trial used 58 pens each containing approximately 500 broilers (Ross 308) at a stocking density of 40 kg/m^2^. The environmental enrichments were: roughage, vertical panels, straw bales, elevated platforms (5 and 30 cm), increased distances between feed and water (7 and 3.5 m) and stocking density reduced to 34 kg/m^2^, plus a control group. Mortality was recorded daily and feed intake and weight weekly. Footpad dermatitis was assessed on day 35. Only one intervention improved financial performance (3.5 m between feed and water) above the control, suggesting that most environmental enrichment would have a negative financial impact due to the additional intervention costs, unless consumers were willing to pay a price premium.

## 1. Introduction

It has been suggested that the inactivity of fast-growing broilers reared in intensive indoor systems negatively impacts leg health, for example leading to gait problems, leg deformities and footpad dermatitis (FPD) [1,2,3]. To address these welfare problems some recent studies have examined the possibility of improving leg health by increasing bird activity by means of providing environmental enrichment. Examples of types of environmental enrichment tested include the provision of perches [4,5,6], placing hay or straw bales into housing [7,8], introduction of sand trays, vegetable materials, hanging objects [5], a dust-bathing substrate, and mirrors [8,9]. All of these enrichments involve changing the nature of the rearing environment to increase its complexity. While some understand environmental enrichment purely in these terms, Newberry [10] has argued that true enrichment must have ‘functional relevance’, i.e., it must bring about some meaningful and positive change in bird behaviour or welfare. In addition, Van de Weerd and Day [11] add further requirements to Newberry’s criteria of environmental enrichment, stating that enrichment must be both practicable and economically beneficial. Broadly speaking, the higher the investment and management costs associated with environmental enrichment, the lower the likelihood that producers will adopt them, thereby reducing their transformative power. Additionally, the less positive the impact of the enrichment on productivity, the lower the likelihood that increases in financial returns will be able to offset higher production costs, and the lower the likelihood of uptake [11]. 

The primary aim of past studies of environmental enrichment has been to test Newberry’s criterion of functional relevance, i.e., their efficacy in driving health improvements. They have done this using a range of different health and welfare measures. For example, Ohara et al. [6] found that the provision of hay bales and perches increased bird activity levels (more standing and moving) in both male and female broilers and reduced the severity of footpad dermatitis (FPD) in female birds. On the other hand, Bailie et al. [7] found no effect of either straw bales or natural light on FPD, while Sans et al. [5] found no effect on FPD of providing perches, sand tray, kale, green cabbage and hanging objects. 

Thus far, none of the studies identified in the literature have tested the economic effect of provision of environmental enrichment. This limitation is of particular relevance because it is well understood in the broiler industry that adoption of environmental enrichments has cost implications and may put additional time demands on stockmen. This study, therefore, sets out to address this limitation by exploring the financial implications of eight types of environmental enrichment (Table 1). The data used in this analysis was derived from a larger study reported in several earlier studies [12,13,14,15]. These previously published studies investigated the effect of the same eight types of environmental enrichment on a range of health and welfare measures, fearfulness, learning ability and activity levels. The purpose of this study is to assess the economic impacts of these enrichments, accounting for the impact of enrichment on production parameters and FPD, and accounting for the direct costs of the enrichment.

## 2. Materials and Methods

### 2.1. Subjects and General Procedures

All procedures involving animals were approved by the Danish Animal Experiments Inspectorate in accordance with the Danish Ministry of Justice Law number 382 (June 10, 1987) and Acts 333 (May 19, 1990), 726 (September 9, 1993), and 1016 (December 12, 2001). The birds were visually inspected daily by trained staff. If any bird was seen in obvious distress (e.g., unable to stand on both legs or walk), it was immediately removed from the experimental room and culled by a percussive blow to the head to make the bird unconscious followed by cervical dislocation.

Day-old, mixed-sex, Ross 308 broiler chicks were acquired from the commercial hatchery DanHatch A/S, Sønderborg, Denmark and reared in two identical 10.7 × 16.6 m rooms in the same building at experimental facilities at Aarhus University. Each room contained five 29.8 m^2^ pens. On the day of placement, the light schedule was programmed for 23 h of light, then reduced by one hour each day until an 18L:6D split was reached on day 6—this was maintained throughout the experiment. Light intensity, as measured at chicken height in three places in each pen (Elma 1335, America A/S, Thisted, Denmark), was approximately 27.5 lx. Natural daylight was not provided to the birds during the study. Rearing conditions were matched to commercial practice as closely as possible. The feeding regime was designed by a local commercial feed company (DLG, Tjele, Denmark). Feed was available ad libitum in round feeders (1.61 cm of feeder space/bird). The number of broilers/nipple drinker was 11.7 (range 11.6–11.8). A 4-cm layer of wood shavings covered the floor in each pen. Flocks were maintained at a maximum stocking density 40 kg/m^2^, except where this rate was reduced as part of the treatment. The stocking density was calculated based on the desired target slaughter weight of 2.2 kg/bird and took into account the area of the pen occupied by the enrichment objects (see Section 2.2). All flocks were slaughtered at 35 days of age. 

The study consisted of eight experimental groups and one control group (Table 1). The study was performed in six blocks, each of 10 pens. Experimental groups were randomly assigned to the pens in each block, with one experimental group assigned twice in each block. Following random allocations, the treatments were balanced across the two adjacent rooms to preclude any confounding effects of the rooms. There were 497 birds in the control pens and between 497 and 422 birds in the experimental pens, depending on treatment (see Table 1). During the trial, a minor flooding incident affected Block 1, and so two pens from that block had to be excluded from the data, resulting in usable data from 58 pens across the six blocks. A summary description of each of the treatments can be found in Table 1.

### 2.2. Experimental Treatments

Enrichments A and B involved increasing the distance between the feeders and the drinkers, i.e., from 1.5 m to 7 m and 3.5 m, respectively. In all other experimental groups and in the control the distance between feeders and drinkers was 1.5 m. In enrichment A, because the birds grew to occupy most of the floor space, the distance between drinkers and feeders was reduced to 1.5 m from day 22 until slaughter, to maintain adequate food and water intake. In enrichment B, the distance between feed and water was kept at 3.5 m throughout the life of the flock. Enrichment C included the addition of a lifetime supply of high-fibre maize roughage feed supplement to the diet, with this available ad libitum in three circular pans (ø 0.4 m), distributed evenly across the pen. The pans were regularly topped up to ensure they were never empty. In enrichment D, five opaque vertical panels (60 × 60 cm) were placed in the central area of the pens, with an even distribution. In enrichment E, three straw bales of 42 × 48 × 122 cm (height × width × length) were evenly distributed across the pen. Both panels and straw bales were present upon placement of the chicks and were not exchanged during the lifetime of the flock. In enrichments F and G, elevated platforms made of perforated plastic slats (5.4 × 0.6 m) which birds could easily access and occupy were added to the pens. In enrichment F, the platform was mounted at a height of 30 cm above the bedding and included two access ramps at an incline of 14.5° for ease of access. The area underneath the platform was fenced off and not accessible to the birds. In enrichment G, the height of the platforms was 5 cm above the bedding and did not include access ramps. Enrichment H consisted of a reduction in maximum stocking density from 40 kg/m^2^ to 34 kg/m^2^. Other than stocking density, the conditions in enrichment H were the same as in the control group (I).

In the control group (I), birds were housed under commercial-like conditions without access to environmental enrichment. The maximum stocking density was 40 kg/m^2^ and the distance between feed and drinking nipples was 1.5 m. When calculating the flock size/pen to achieve 40 kg/m^2^, the floor area occupied by the enrichment objects was subtracted from the total floor area of the pen. This was in accordance with Danish and European regulations [16,17], specifying that stocking density must be calculated on usable area covered by litter and accessible to the chickens at any time. Therefore, the floor area occupied by the straw bales and the elevated platforms did not count as net floor area. To account for the resulting difference in flock size/experimental group, the number of drinking nipples and feeding space/bird was also adjusted to preclude any confounding effects of altered competition for resources.

### 2.3. Data Collection

The amount of feed provided to the broilers every day was measured and totaled for the week for each pen. At the end of each week, the amount of feed left in the feeders was measured and this value deducted from the initial amount of feed, to arrive at the amount consumed in each pen that week. This value was then divided by 7 (days) and divided by the number of surviving birds in that week, to obtain average feed intake/bird/d in that week. Average daily feed consumption over the life of the birds was obtained by summing the average daily feed intakes calculated for each of the five weeks and dividing by five. Mortality was monitored every day. Any birds found dead, or which had to be culled, were counted and the number summed/w and then for all five weeks. From this total, and the number of chicks placed, the cumulative mortality rate for each pen over the life of the trial was calculated. The weight of 100 broilers/pen was measured on days 0, 7, 21, and 35. The average start weight was deducted from the average broiler weight/pen on day 35. This resulting value was then divided by 35 d to arrive at the average daily weight gain for each pen. Feed conversion ratio, for each pen, was calculated by dividing lifetime average feed intake by the lifetime average weight gain. Finally, footpad dermatitis was assessed on a scale from 0 to 2 on day 35 of age. For more information on footpad dermatitis see [12].

### 2.4. Calculating Financial Impacts

To estimate the financial impacts of the eight environmental enrichments, a standard broiler enterprise cost model was constructed (See supplementary material Appendix A), based on average broiler production costs data for Denmark for 2013 [18]. These costs data were available on a per kg live weight basis and were applied first to the control group (Enrichment I). In applying these costs, an adjustment was made for the fact that the average slaughter weight of the birds produced in the trial was higher than the published national average for Denmark in 2013 [18], i.e., 2417.88 g/bird for the trial compared with the national average of 2300 g. To reflect the fact that the trial produced bigger birds/unit of input, non-feed costs/kg live weight were adjusted downwards by the ratio of the trial to national average slaughter weights. In the case of feed costs, a separate adjustment was made to account for the better feed conversion ratio (FCR) in the trial control group compared to the national average FCR for Denmark [18]. The 2013 market price of broiler outputs, on a/kg live weight basis, was derived for Denmark from Eurostat [19], with 2013 being the most recent year for which data were available at the time of analysis. 

To calculate the financial impacts of each enrichment, the control group costs were adjusted on the basis of the differences in slaughter weight, mortality and feed conversion ratio between the control and each enrichment, even where these changes were not statistically significant. It was assumed, for simplicity sake, that birds that died during the trial consumed, on average, half of their potential total life feed requirement. Therefore, feed costs in each enrichment group were adjusted by half of the difference in mortality percentage between the control and each of the enrichment groups. 

Additionally, the direct cost of implementing each type of environmental enrichment was accounted for (see Table 2) by adding these costs as a separate cost category in the cost model (this allowed for greater transparency). These costs were restricted to materials, as no data were available on the management and labour costs required for installation, or any additional ongoing labour costs arising from the resultant changes to the physical environment (Appendix B). Where enrichment costs involved lump-sum capital expenditures for durable materials, these costs were spread evenly over an estimated 20-year product life, in current price terms. The profit measure used throughout the analysis was net margin, estimated either on a per kg liveweight or per enterprise basis. Net margin was defined as market returns (sales of broiler product), less all variable and fixed costs. Finally, for the enrichment types that required a reduction in flock size (reduced stocking density, straw bales and platforms) an increase in fixed costs (Housing, labour, general overheads)/kg liveweight was calculated to reflect the loss of throughput (birds produced/m^2^).

In addition to impacting production costs, the enrichments also impacted revenues by affecting FPD severity which, in Denmark, influences carcass downgrades at the slaughterhouse. The rate of this price change, under normal commercial practice (Denmark), is calculated as follows. A sample of birds from each flock is inspected for FPD at the slaughterhouse and each bird is graded using a 3-point scale, i.e., 0 (no ulcerations), 0.5 (less serious ulcerations), or 2 (severe ulcerations) [20]. The price adjustment is based on an uplift of 0.83% for each individual score of zero and a reduction in price of 4.99% for each score of 2 [21]. Using this approach, price adjustments were calculated for this study. The distribution of FPD scores presented by Tahamtani et al. [12] in each enrichment was used to calculate a percentage change in the producer price compared to the control.

### 2.5. Statistical Analysis 

All data analysis was undertaken using the analytics software SAS version 9.4 [22]. 

The effects of enrichment on mortality, lifetime average feed consumption, lifetime average weight gain, feed conversion ratio and slaughter weight were analysed using the ‘mixed’ procedure with block as a random effect. Pen was not included as a random effect, as the physical measure estimates are, in this case, averages for each pen, i.e., pen is the unit of replication. Enrichment was included as a fixed effect. Where significant effects of enrichment were found, post-hoc pairwise comparisons of enrichments were performed using the Tukey HSD (Honestly Significant Difference) procedure, using LS Means. 

As reported above, a spreadsheet-based, standard broiler enterprise cost model, accounting for all market-based revenues, i.e., sales of birds and both variable and fixed costs (Appendix A), was used to cost the changes to production parameters, averaging over block and pen, to yield a single net margin value for each enrichment and control group. Because these net margin values have been generated partly from data that have no standard deviation values, such as cost of chickens and revenue/bird, it was not possible to perform statistical testing of the significance of group differences.

## 3. Results

### 3.1. Impact of the Enrichments on Production Parameters

Table 3 presents the effect of the different enrichments on mortality, lifetime average feed consumption, lifetime average weight gain and feed conversion ratio. There was no significant effect of enrichment on feed conversion ratio (F_8,44_ = 1.31; *p* = 0.26) or total mortality (F_8,46_ = 0.69; *p* = 0.7). There was, however, a significant effect of enrichment on daily feed consumption (F_8,44_ = 2.17; *p* = 0.048), although there were no significant differences between any of the enrichments and the control group on this measure; the only significant difference being that enrichment C had a significantly lower rate of feed consumption than enrichment F (df = 44, *t* = −3.83, *p* = 0.010). There was a very close to significant effect of enrichment on terminal bird weight (F_8,44_ = 2.15; *p* = 0.051), but post-hoc tests revealed that none of the enrichments generated any significant differences from the control group. The only significant effect was that enrichment C yielded a lower slaughter weight than enrichment F (df = 44, *t* = 3.64, *p* = 0.018). There was also a significant effect of enrichment on daily weight gain (F_8,44_ = 2.17; *p* = 0.048), but no individual enrichment resulted in a rate of weight gain that was different from the control. The one significant enrichment effect was the lower rate of weight gain in enrichment C, compared with enrichment F (df = 44, *t* = −3.66, *p* = 0.017). 

### 3.2. Financial Impacts of the Interventions 

The effects of enrichment on FPD are presented by Tahamtani et al. [12]. Here, we used the differences in FPD between the enrichments and the control to estimate changes in producer price. As Table 4 shows, declines in average FPD score below that of the control (i.e., improvements in footpad condition) resulted in the payment of a modest price premium in two enrichment groups, with the highest being a price uplift of 0.43% in enrichment group F, followed by 0.17% for enrichment group G. In contrast, enrichments A, B, C, E, and H incurred a small price penalty because of inferior FPD scores compared to the control, with the largest penalty being −0.47% in enrichment group A. 

Table 5 shows that some of the enrichments resulted in a lower net margin on a per kg live weight basis than the control group, while others showed a small increase. The largest falls in net margin were evident for enrichment C (−4.29%; roughage) and for enrichment E (−3.98%; straw bales). Only two enrichments, B and H, resulted in improvements in net margin compared to the control. The largest improvement in net margin (+0.71%) occurred for enrichment B (Distance between feed and water—3.5 m). However, enrichment H had a loss of throughput of 15% as a result of the lowered stocking density, thereby cancelling out the increase in net margin. Furthermore, due to reduced flock size enrichments E (straw bales), F (30 cm platforms), and G (5 cm platforms) also had losses of throughput of 3%, 12.1%, and 12.1 %, respectively.

## 4. Discussion

The current study presents results on the effects of eight different types of enrichment on production parameters and financial performance in broiler production. Birds in enrichment group C (maize roughage) had lower feed consumption compared to enrichment group F (30 cm elevated platforms). This also led to significant differences in daily weight gain and slaughter weight between this same pair of enrichments. A likely reason for this effect is that birds provided with maize eat less of the ad libitum food as they eat the nutritionally less rich maize roughage instead (although bird consumption of the roughage was not directly measured, this was evident from the requirement to regularly top up the roughage pans). Another possible explanation is that access to roughage promotes foraging behaviour, increasing the levels of physical activity and, thus, energy expenditure. Past studies showed that scattering feed in litter, in an effort to increase activity levels, can reduce terminal body weight by as much as 13% [23]. Indeed, in the current study, broilers provided with roughage were more active than either those housed with elevated platforms or with increased distances between feed and water at 20 and 27 days of age [13]. 

While only enrichment C (roughage) and F (30 cm platforms) had statistically significant impacts on production parameters, it is relevant to examine the financial impact of all types of enrichment for two reasons. First, most types of environmental enrichment cost money to deploy, both in the form of investment capital and higher management costs. Second, the effects of enrichment on production factors that are not statistically significant can still have notable impacts, in terms of percent change in net margin, as changes in production costs and output losses are amplified here. Enrichment B (3.5 m between feed and water) and H (reduced stocking density) were the only enrichments that succeeded in improving net margin on a per bird basis above that of the control group. The more beneficial of the two was enrichment B (3.5 m between feed and water) which generated a 0.86% increase in net margin above that of the control group, amounting to € 21,345 additional profit/year, for a 500,000 bird/year unit. This positive impact is due to a slightly reduced feed consumption and better FCR and due to the fact that no additional costs were required to implement an increased distance between feed and water. Likewise, Balog et al. [24] found an improvement in FCR in birds challenged to exercise more compared to controls, which the authors attributed to decreased lameness and improved circulation. Nevertheless, any increase of the distance between feed and water should be carefully considered and monitored as it may impair the accessibility of these resources by the birds as they grow and become less mobile.

The largest fall in net margin across the interventions (−4.29%) was observed for enrichment C (roughage). This fall in net margin was a result of increased production costs/kg, together with a very slight decrease in the price paid for each bird, due to elevated carcass downgrades, resulting from a small increase in FPD severity. The elevated production costs resulted from reduced production volumes, and the reduced production volumes were caused by a combination of reduced feed consumption, lower slaughter weight and a slight increase in bird mortality. As previously described, this was likely due to reduced appetite for the broiler feed, due to the consumption of maize roughage, with its lower nutritional value and protein content [25].

The second largest numerical fall in net margin (−3.86%) was evident for enrichment E (straw bales), resulting from both increased production costs and a decline in the average carcass price, due to elevated FPD severity. Previous studies have identified straw, particularly unchopped straw, as a risk factor for the development of footpad dermatitis, due to its effect on the moisture content of the litter [26,27,28]. Production costs/kg live weight rose under this enrichment due to lower rates of feed consumption, leading to reduced average daily weight gain and slaughter weights. FCR was slightly improved, but mortality was also elevated. This result contradicts the findings of Ohara et al. [6], who found a positive enrichment effect of straw bales and perches on FPD in female birds, but not males. No separate analysis by sex was undertaken in this study. The introduction of straw bales into broiler houses is one of the standard approaches to environmental enrichment recommended by the UK’s Royal Society for Prevention of Cruelty to Animals [29]. It should be pointed out that straw bales were added to pens in this trial at a higher rate (one bale for every 161 birds) than under the UK’s Royal Society for the Prevention of Cruelty to Animals (RSPCA) guidelines (one bale/667 birds). This higher rate of availability would perhaps have acted to exaggerate the effect of the bales, for example on FPD. 

The full financial impacts of some of the enrichments were only revealed when changes to flock size were factored in. For example, in the case of enrichment H the maximum stocking rate was cut from 40 kg/m^2^ to 34 kg/m^2^, thereby reducing the flock size and resulting in a cut of 15% in throughput/pen. However, there were also losses of throughput in the cases of enrichments E (−3%), F (−12.1%) and G (−12.1%). In these cases, while the density of birds/m^2^ of floor space was not reduced, the amount of floor space itself was reduced, leading to lower throughput/pen. To give an example of the consequence of the loss of throughput, in enrichment H (low stocking density) the loss of throughput of 15% would result in a loss of €36,513/year for a 500,000 bird/year unit, this is equivalent to a net margin fall of 2.5% for each 1 kg/m^2^ cut in stocking rate. This projected loss of net margin under reduced stocking rates is supported by a number of other studies [29,30]. Verspecht et al. [30] combined data from three trials in Belgium over the period 1996–2008 with varying levels of reductions in stocking rates, and estimated an average loss of net margin of 3.1% for every 1 kg/m^2^ reduction in stocking density. Utnick-Banaś et al. [31] estimated a 5.3% reduction in net margin for each 1 kg/m^2^ cut in stocking rates for a small sample of broiler producers in Poland, averaged over the years 2009–2011. Previous studies have found significant health and welfare benefits from reduced stocking density, for example Hall [32] and Knierim [33] both showed that locomotion and foraging activities increase as stocking density decreases, while Hall [32] also showed that lowering stocking density improved leg health. Besides such direct health and welfare benefits, Meluzzi [34] demonstrated that lower stocking density may be associated with improved litter quality, thereby yielding potential indirect health and welfare benefits, for example in terms of reductions in FPD. While these health and welfare improvements may well yield some financial gains, the current study suggests that these would be insufficient to cover the loss of revenues resulting from the loss of throughput. 

Enrichments F and G resulted in loss of throughput because the number of birds was reduced. This was required to maintain stocking densities following loss of floor space resulting from the introduction of the elevated platforms. However, while platforms at these relatively low heights do not count as floor area according to Danish regulation (i.e., birds cannot access the floor space beneath the platforms), other countries might have different regulation which does not require reductions in flock size when enrichment objects are added to floorspace. However, even discounting for the loss of throughput, the current study showed that net margin/kg of output would still be reduced when platforms were provided.

It might be assumed, based on the scale of the financial losses resulting from some of these enrichments strategies, that broiler producers would only adopt them if compelled to do so by regulation. However, if their efficacy in delivering welfare benefits could be established in the mind of the consumer, a market alternative to regulation may present itself. Taking the most financially disadvantageous of the interventions by way of illustration, i.e. enrichment H (reduced stocking density from 40 kg/m^2^ to 34 kg/m^2^), to compensate producers for the 15% drop in throughput, consumers would need to pay an additional € 0.029/kg live weight (around € 0.07/bird), a price increase of around 3%. This is a relatively modest level of price increase and therefore it is feasible that the market might absorb it. Indeed, the market already absorbs the even higher costs associated with free range systems and also the higher costs of products marketed with welfare credence values, for example enhanced-welfare labels, such as the Beter Leven National Animal Welfare label in the Netherlands and the RSPCA [35] assured label in the UK [36]. All of these assurance schemes require lower stocking densities, together with a number of other environmental and management changes to conventional practice. 

Finally, it should be pointed out that while an experimental trial, such as is reported here, can give indications of types of environmental enrichment that might be beneficial in terms of bird health and welfare, or financial performance, these enrichment strategies need to be tested under commercial conditions with the variation and practical constrains that exist there. 

## 5. Conclusions

The findings of this study suggest that care must be taken when selecting, in commercial farming practice, environmental enrichment to improve bird welfare. Ideally, enrichment should be both practical, profitable and improve welfare or health. In the study, increasing the distance between food and water was the only enrichment that did not require any additional costs and an increased distance of 3.5 m between food and water was also the only enrichment that was profitable compared to the control. These results demonstrate the need for a price premium if the provision of environmental enrichment is to be profitable, especially in cases were the environmental enrichment itself is costly. However, profitability is merely one of several factors that can be affected by environmental enrichment, and factors such as health, welfare, and behaviour should be taken into account when selecting which types of enrichment to further investigate. 

## Figures and Tables

**Table 1 animals-10-00378-t001:** Experimental groups, flock size/pen and total number of pens for each enrichment across all six blocks.

Experimental Group Code	Experimental Group	Flock Size/Pen	No. of Pens
**A**	Distance between feed and water—7 m	497	6
**B**	Distance between feed and water—3.5 m	497	6
**C**	Maize roughage	497	7
**D**	Vertical panels	497	6
**E**	Bales of straw	482	7
**F**	Platform at 30 cm height and access ramps	437	6
**G**	Platform at 5 cm height, no access ramps	437	6
**H**	Lower stocking density (34 kg/m^2^)	422	6
**I**	Control	497	8

**Table 2 animals-10-00378-t002:** Direct costs of each enrichment type applied in the study.

Cost Source	A	B	C	D	E	F	G	H	I
Cost of materials (€ cents/kg)	0	0	0.492	0.114	0.302+	0.433+	0.416+	0	0
Cost of reduced flock size (% increase in fixed cost/kg)					3.1%	13.7%	13.7%	17.6%	

Enrichments: A—7 m distance between feed and water; B—3.5 m distance between feed and water; C—maize roughage; D—vertical panels; E—straw bales; F—30 cm elevated platform; G—5 cm elevated platform; H—max. stocking density reduced from 40 kg/m^2^ to 34 kg/m^2^; I—control.

**Table 3 animals-10-00378-t003:** Mean ± standard deviation of lifetime average feed consumption, average slaughter weight, feed conversion ratio, lifetime average weight gain and total mortality across enrichments.

Enrichments	Feed Consumption (g/bird/day)		Average Slaughter Weight (g)		Feed Conversion Ratio		Weight Gain (g/bird/day)		Total Mortality (%)	
	Mean	Std Dev	Mean	Std Dev	Mean	Std Dev	Mean	Std Dev	Mean	Std Dev
**A**	98.1	6.4	2410.5	104.2	1.45	0.07	67.7	3.0	2.65	0.65
**B**	97.8	5.7	2410.3	54.5	1.45	0.08	67.7	1.5	2.48	1.09
**C**	95.8 ^ab^	5.6	2370.2 ^a^	88.6	1.44	0.07	66.5 ^a^	2.5	2.76	1.29
**D**	99.9 ^a^	6.1	2417.3	77.7	1.47	0.06	67.9	2.3	2.78	0.91
**E**	97.9	4.7	2409.2	40.4	1.45	0.07	67.6	1.1	2.79	0.88
**F**	99.5 ^b^	6.6	2462.6 ^a^	91.6	1.44	0.06	69.2 ^a^	2.6	2.52	1.11
**G**	98.0	6.1	2419.5	68.3	1.44	0.09	67.9	2.0	1.87	0.89
**H**	98.2	7.1	2448.5	83.8	1.42	0.09	68.7	2.4	2.29	0.61
**I**	99.1	7.0	2417.9	101.2	1.46	0.07	67.9	2.9	2.35	0.82

Enrichments: A—7 m distance between feed and water; B—3.5 m distance between feed and water; C—maize roughage; D—vertical panels; E—straw bales; F—30 cm elevated platform; G—5 cm elevated platform; H—max. stocking density reduced from 40 kg/m^2^ to 34 kg/m^2^; I—control. ^a–b^—Different superscript letters within a column indicate significantly different values (*p* ≤ 0.05).

**Table 4 animals-10-00378-t004:** Percent change in the average producer price resulting from the average changes to FPD score in each enrichment group compared to the control.

Impact	Enrichments							
	A	B	C	D	E	F	G	H
Change in producer price (%)	−0.47	−0.17	−0.19	0.00	−0.44	0.43	0.17	−0.05
Change in average FPD severity score	+0.15	+0.08	+0.07	−0.04	+0.27	−0.27	−0.17	−0.03

Enrichments: A—7 m distance between feed and water; B—3.5 m distance between feed and water; C—maize roughage; D—vertical panels; E—straw bales; F—30 cm elevated platform; G—5 cm elevated platform; H—max. stocking density reduced from 40 kg/m^2^ to 34 kg/m^2^. Note: Because of the disproportionate weighting given to FPD scores of 2 compared to scores of zero, a small increase in the number of scores of 2 can result in a negative change in average price paid, even in cases when there is a small improvement in average FPD score, as is the case with Enrichment H.

**Table 5 animals-10-00378-t005:** Financial assessment of the control and interventions, plus financial impacts of interventions on production cost/kg live weight.

Enrichments	A	B	C	D	E	F	G	H	I
	€ cents/kg	€ cents/kg	€ cents/kg	€ cents/kg	€ cents/kg	€ cents/kg	€ cents/kg	€ cents/kg	€ cents/kg
Revenue—Live bird	97.42	97.71	97.69	97.88	97.49	98.30	98.30	97.84	97.88
Variable costs									
Chicks	14.29	14.26	14.63	14.35	14.40	14.05	14.20	14.13	14.29
Feed	45.98	45.94	45.67	46.64	46.01	45.62	45.47	44.84	46.23
Other variable costs	7.55	7.54	7.74	7.58	7.61	7.43	7.51	7.47	7.55
Direct intervention costs	0.00	0.00	0.49	0.11	0.30	0.43	0.42	0.00	0.00
Fixed costs									
Labour	3.46	3.45	3.54	3.47	3.59	3.87	3.91	4.02	3.46
Housing	5.00	5.00	5.13	5.03	5.20	5.60	5.66	5.82	5.01
General overheads	0.82	0.82	0.84	0.82	0.85	0.92	0.93	0.95	0.82
Net margin (€ cents / kg)	20.32	20.71	19.65	19.87	19.51	20.38	20.21	20.60	20.53
Change in net margin from control (%)	−1.04	0.86	−4.29	−3.20	−4.97	−0.71	−1.57	0.34	

Enrichments: A—7 m distance between feed and water; B—3.5 m distance between feed and water; C—maize roughage; D—vertical panels; E—straw bales; F—30 cm elevated platform; G—5 cm elevated platform; H—max. stocking density reduced from 40 kg/m^2^ to 34 kg/m^2^; I—control.

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
