# Peer review of "The Productivity and Financial Impacts of Eight Types of Environmental Enrichment for Broiler Chickens"

_animals, 2020, doi:10.3390/ani10030378_

Round 1

Reviewer 1 Report

Line 84: This is the only reference to sex. I would expect that in the end some check is done on variation in male-female ratio per pen, as this can influence average body weight per pen. I can’t find it anywhere..

Line 93 -130: The order of the text is a bit strange:

  • Line 93: I miss an explanation of how this was calculated, how is the usable space determined? From table 1 in line 106 I can derive some, but not all
  • Line 106: I can’t find statements how usable surface is calculated.
  • Finally in line 130 I can find this explanation.

I would suggest to put the text starting at line 130 before  line 93.

Line 234: the treatment letter C is missing

Line 254: up to this I suppose the text belongs to table 4, so after this there should be a white line

Line 291: reference author is missing

Line 355: the last sentence is no conclusion, but a discussion point

Author Response

Line 84: This is the only reference to sex. I would expect that in the end some check is done on variation in male-female ratio per pen, as this can influence average body weight per pen. I can’t find it anywhere..

Thank you for the comment. Unfortunately we were not able to assess the sex of the individual birds and so we did not perform any analysis on sex variations.

***********************************************************

Line 93 -130: The order of the text is a bit strange:

  • Line 93: I miss an explanation of how this was calculated, how is the usable space determined? From table 1 in line 106 I can derive some, but not all
  • Line 106: I can’t find statements how usable surface is calculated.
  • Finally in line 130 I can find this explanation.

I would suggest to put the text starting at line 130 before line 93.

 We have added the following sentence at Line 96 to increase clarity:

“The stocking density was calculated based on the desired target slaughter weight of 2.2 Kg/bird and took into account the area of the pen occupied by the enrichment objects (see section 2.2)”

***********************************************************

Line 234: the treatment letter C is missing

We have added the letter C.

***********************************************************

Line 254: up to this I suppose the text belongs to table 4, so after this there should be a white line

We have added a paragraph line to more clearly separate the table footnotes from the main text (now Line 263).

***********************************************************

Line 291: reference author is missing

Reference author name has been added to the text (which is now at Line 301).

***********************************************************

Line 355: the last sentence is no conclusion, but a discussion point

Agreed: this sentence has been reworded slightly and moved to the end of the Discussion section (now Line 373).

Reviewer 2 Report

This is a very well written manuscript, which presents clear and relevant results on a topic that is ever growing in interest, both in the eyes of the industry and the society. I only have minor comments to the manuscript:

L237: the authors mention several times that the birds ate the roughage, and speculate that this would impact the results. Was the amount of roughage given to the birds weighed or recorded in any way? This information would be helpful, and reduce the need for speculation. 

L285: treatment H (reduced density) improved net margin per bird. The authors could elaborate on other potential positive effects of reduced density on health, welfare and factors such as effect on litter quality presented in earlier papers from this and other studies.   

Author Response

L273: the authors mention several times that the birds ate the roughage, and speculate that this would impact the results. Was the amount of roughage given to the birds weighed or recorded in any way? This information would be helpful, and reduce the need for speculation. 

Thank you for the comment. We are positive that the birds consumed the roughage, as this was replenished every day. The amount of roughage consumed was only roughly measured so to ensure that the birds would receive a portion large enough to be ad libitum. Unfortunately, we did not keep detailed measurements of the roughage provided and therefore cannot provide these consumption statistics. We have added a clarifying statement to the paper at Line 283.

************************************************************

L285: treatment H (reduced density) improved net margin per bird. The authors could elaborate on other potential positive effects of reduced density on health, welfare and factors such as effect on litter quality presented in earlier papers from this and other studies.   

We have added the following text to the Discussion at Line 343:

‘Previous studies have found significant health and welfare benefits from reduced stocking density, for example Hall [32] and Knierim [33] both showed that locomotion and foraging activities increase as stocking density decreases, while Hall [32] also showed that lowering stocking density improved leg health. Besides such direct health and welfare benefits, Meluzzi [34] demonstrated that lower stocking density may be associated with improved litter quality, thereby yielding potential indirect health and welfare benefits, for example in terms of reductions in FPD. While these health and welfare improvement may well yield some financial gains, the current study suggests that these would be insufficient to cover the loss of revenues resulting from the loss of throughput. .’

Reviewer 3 Report

The article with the title „The productivity and financial impacts of eight types of environmental enrichment for broiler chickens“ provides important and interesting information on the costs of different implemented environmental enrichments in broiler husbandry. The material and methods section seems coherent and the study design and statistical evaluation appropriate.

Although here are some minor remarks:

Line 21 ff: Increasing a distance between feed and water does not seem like a proper „enrichment“ for the enhancement of broiler welfare. German law requires for example that all broilers must have equally access to feeders. This cannot be ensured for all animals in a large flock or in flocks with health problems when distances are increased. This aspect will not be convincing for consumers that are willing to pay more for animal welfare as in label products. The authors might want to discuss this.

Line 89: How was light intensity measured? Lux measuring device (type)? Please give more information about location, number and results of measurements. There are lighter and darker spots in a barn. Measuring illuminance has to be seen as snapshot measurement if not measured coninuously and can change in the course of the day when daylight is provided in a barn. Was there at some point daylight provided in the study?

Lines 120/121/134: If non perforated slats are used for example in Germany floor space can be credited and no reduction of animals is necessary. You might want to pick up this aspect in your discussion/calculation. Elevated platforms are seen as a „real“ enrichment for the birds compared to increase of distance to food an water, if managed properly.

Line 234: C-maize roughage. The C is missing here.

Line 364: culled by cervical dislocation. I hope with prior stunning to it. How are emergency killings embodied in the Denmark law?

Author Response

Line 21 ff: Increasing a distance between feed and water does not seem like a proper „enrichment“ for the enhancement of broiler welfare. German law requires for example that all broilers must have equally access to feeders. This cannot be ensured for all animals in a large flock or in flocks with health problems when distances are increased. This aspect will not be convincing for consumers that are willing to pay more for animal welfare as in label products. The authors might want to discuss this.

Thank you. We agree that increasing the distance between the feed and the water might compromise the birds’ access to these resources as they grow. For this reason, we returned the distance between feed and water of treatment A to the control distance (i.e. from 7m to 1.5m) on day 22 of age. We have added the following sentence to the discussion at Line 304:

“Nevertheless, any increase of the distance between feed and water should be carefully considered and monitored as it may impair the accessibility of these resources by the birds as they grow and become less mobile.”

*************************************************************

Line 89: How was light intensity measured? Lux measuring device (type)? Please give more information about location, number and results of measurements. There are lighter and darker spots in a barn. Measuring illuminance has to be seen as snapshot measurement if not measured continuously and can change in the course of the day when daylight is provided in a barn. Was there at some point daylight provided in the study?

Thank you. We have added information (at Line 89) on the lux measurer and how the light intensity was measured. We have also mentioned that natural daylight was never provided in the study.

*************************************************************

Lines 120/121/134: If non perforated slats are used for example in Germany floor space can be credited and no reduction of animals is necessary. You might want to pick up this aspect in your discussion/calculation. Elevated platforms are seen as a „real“ enrichment for the birds compared to increase of distance to food and water, if managed properly.

Thank you for this intelligence. We have added the following text to the discussion at Line 351 to make this point apparent.

Thank you for this intelligence. We have added the following text to the discussion at Line 351 to make this point apparent.

Enrichments F and G resulted in loss of throughput because the number of birds was reduced. This was required to maintain stocking densities following loss of floor space resulting from the introduction of the elevated platforms. However, while platforms at these relatively low heights do not count as floor area according to Danish regulation (i.e. birds cannot access the floor space beneath the platforms), other countries might have different regulation which does not require reductions in flock size when enrichment objects are added to floorspace. However, even discounting for the loss of throughput, the current study showed that net margin per kg of output would still be reduced when platforms were provided.

************************************************************

Line 234: C-maize roughage. The C is missing here.

The missing letter C has been added.

************************************************************

Line 364: culled by cervical dislocation. I hope with prior stunning to it. How are emergency killings embodied in the Denmark law?

In Denmark, stunning is also required prior to cervical dislocation, and emergency killing during our project of course followed this procedure. At Line 396 we changed the sentence to:

…”it was immediately removed from the experimental room and culled by a percussive blow to the head to make the bird unconscious followed by cervical dislocation.”